# Illegitimate Tasks, Negative Affectivity, and Organizational Citizenship Behavior among Private School Teachers: A Mediated–Moderated Model

**Nessrin Shaya [1], Laila Mohebi [2],\* 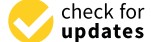, Rekha Pillai [3] and Rawan Abukhait [4]**

[1] College of Humanities and Sciences, Ajman University, Ajman P.O. Box 346, United Arab Emirates; n.shaya@ajman.ac.ae
[2] College of Humanities and Social Sciences, Zayed University, Dubai P.O. Box 19282, United Arab Emirates
[3] Faculty of Business and Law, British University in Dubai, Dubai P.O. Box 345015, United Arab Emirates; rekha.pillai@buid.ac.ae
[4] College of Business Administration, Ajman University, Ajman P.O. Box 346, United Arab Emirates; r.abukhait@ajman.ac.ae
\* Correspondence: laila.mohebi@zu.ac.ae

**Abstract:** Social sustainability has gained popularity over the last decade, with a growing body of research calling for researchers to focus on the personal-level determinants of employee satisfaction and well-being in the pursuit of social sustainability. By using negative affectivity as a mediating mechanism and gender and passive leadership as moderators, this study examines a novel sequential mediation–moderation model that explores the relationship between unreasonable tasks and teachers' Organizational Citizenship Behavior (OCB). It employs the Conservation of Resources (COR) and Stress as Offense to Self (SOS) paradigms as a comprehensive theoretical framework for organizational stressors and organizational behavior. A total of 415 matched questionnaire responses were collected from private school teachers in the UAE. Confirmatory factor analysis (CFA) is conducted using AMOS 20, hierarchical linear modeling (HLM) is utilized to verify the causal and moderation hypotheses, and the resulting moderated mediated conceptual model is evaluated by employing Hayes PROCESS analysis. Results demonstrate the effects of illegitimate tasks on OCB are indirect and statistically significant and are mediated through negative affectivity. The cumulative effect of illegitimate tasks and negative affectivity on OCB is magnified by the moderating effects of passive leadership.

**Keywords:** social sustainability; illegitimate tasks; negative affectivity; passive leadership; organizational citizenship behavior; teachers

## 1. Introduction

The industrial composition of global economies has witnessed radical changes over time, with digital revolutions inducing myriad challenges and complexities to the educational sector work environment and working conditions in the form of the greater complexity of jobs, transition from on-campus to online teaching, unrealistic expectations from supervisors, extended working hours, diminished time for family, and reduced work–life balance [1]. In recent years, organizational researchers have increasingly studied the positive outcomes of thriving at work, which can be interpreted as a state of mental vitality and development [2]. Fostering healthy and highly engaged employees is vital to achieving social sustainability. Embracing employees' behaviors shows positivity towards their organization, high employee engagement, efficient and effective task performance, and greater attention to organizational goals. They are innovative and creative, and they have a strong emotional connection to their organizations [2]. Research confirms that thriving at work is associated with increased job satisfaction, physical and mental health, dedication, creativity, a focus on self-improvement, organizational citizenship, job performance, and a reduction

in employee turnover and burnout [3,4], all of which contribute towards achieving social sustainability goals.

Our study makes a significant contribution to the field of social sustainability, particularly within the educational sector. It delves into the dynamics of teacher workload and stress and their impact on organizational citizenship behavior in private schools. This focus is directly relevant to the pursuit of sustainable educational practices, ensuring the well-being and effectiveness of educators. The theoretical underpinnings of our research, grounded in the Conservation of Resources (COR) and Stress as Offense to Self (SOS) paradigms, provide a comprehensive framework for understanding organizational stressors and behavior. This theoretical contribution enriches the discourse on sustainable practices and well-being in organizational settings, aligning well with increased emphasis on social sustainability. In the context of business, social sustainability involves comprehending the effects of corporations on individuals and society. Social sustainability advocates for organizations to prioritize the fundamental human needs of their employees and actively contribute to positive advancements in society. As such, human rights, ethical labor practices, living conditions, health, safety, wellness, diversity, equity, work–life balance, empowerment, and more are examples of social sustainability performance issues in organizations. The literature has increasingly shown that organizational leadership recognizes the need for social sustainability to ensure future success; however, the majority lack the understanding of how to embed social sustainability into organizational culture and leadership practices. This study addresses this gap, with practical implications for developing sustainable working environments and culture in schools. We uncover the detrimental effect of stressors in the workplace on teachers' mental well-being and the role of leadership in mitigating these effects. We propose strategies for reducing illegitimate tasks, managing teacher workload, and enhancing teachers' helping behaviors. These strategies are crucial for the long-term social sustainability of educational institutions and contribute to the broader sustainability agenda. As such, this research addresses a critical gap in the current understanding of social sustainability in the educational sector. By focusing on teachers' experiences and their impact on organizational behavior, we offer unique insights that are valuable for advancing the broader sustainability discourse.

Myriad studies show that job stress, as a result of the multitude of tasks teachers must perform with enthusiasm, has a significant predictive impact on a teacher's burnout and exhaustion [5], leading to dissatisfaction, absenteeism, and high turnover rates [6]. Illegitimate task (ILT), a recently introduced task-related stressor, has been capturing the attention of scholars and practitioners [2,7–9]. ILT refers to the unnecessary and unreasonable duties that go beyond teacher's requirements and should not be expected from teachers given their professional identity [10]. Accordingly, job stressors have been recognized as one of the most significant issues facing employees in general and teachers specifically [11,12], educational institutions [13], with potential direct detrimental implications on students' achievement [14,15] and overall school performance [16]. Nonetheless, ILTs are not uncommon at work and are often recurrently reported [17]; they may range between two and three times a week in frequency [18], constituting up to 10% of main tasks (such as instruction for teachers) and about 65% of auxiliary tasks [19]. Research has consistently demonstrated the detrimental impact of stressful work environments on the mental health and motivation of employees [18,20,21] due to the emotional exhaustion and burnout experienced. Despite this accumulation of evidence, performance-related output among teachers in the educational sector and associated costs have attracted minimal attention and emerged as a major theoretical and empirical gap. Amongst the few studies examining ILT-employee performance associations [22,23], discussions were either restricted to potential implications on non-education sectors or with minimal reference to teachers specifically. This study addresses this limitation by expanding on existing research and being the first to investigate the significance of detrimental work-related stressors on the Organizational Citizenship Behaviour (OCB) of teachers in relation to students and colleagues. OCB [24] refers to voluntary behavior via helping behavior, self-development,

and sportsmanship [25] exhibited by the individual, which is beyond the job description and non-remunerated but ultimately facilitates and benefits the organization [26]. Additionally, studies with mediating pathways between ILTs and output constructs also remain scarce, with a few papers addressing mediating variables, such as psychological detailing, frustration, self-esteem, and negative affectivity [18,21,27]. This highlights the need for further investigation into the interaction between affective and cognitive variables that play a significant role in the connection between ILT and performance output, such as negative affectivity (NA) [11]. The latter emerges as a direct and immediate response to ILT, with a potential direct impact on performance outcomes. Therefore, by emphasizing NA as an intervening variable that explains the relationship between ILT and OCB towards students and coworkers, this study contributes to this promising area of research.

In an attempt to establish the theoretical underpinnings of the potential direct negative impact of ILT on NA and its potential indirect impact on OCB, the current study utilizes the Conservation of Resources (COR) [28] and Stress as Offence to Self (SOS) [29,30] theories. The COR theory suggests that ILT can drain teachers' personal resources, leading to distress, discontent, and discomfort, which can negatively impact performance outcomes. Concurrently, the SOS theory advocates the notion that threats to self-esteem and self-respect, especially in the form of ILT, are a source of strain and stress [31] or, in other words, NA. We further anticipate that NA aggravates exposure to passive leadership (PL) (a leader who lacks engagement, is uncommunicative, uninterested, and lacks feedback) [32]. PL is characterized by the absence of adequate employee social support, leading to diminished social resources for teachers to effectively process, complete, or handle illegitimate task events [32]. Therefore, in line with the COR theory, the lack of adequate social support from leaders [33] would negatively affect teachers' performance outcomes. Conceptualizing this moderating process can also provide clear guidance to schools on the need to have effective leaders managing teachers' experiences of ILTs that otherwise could not be eradicated. The strength of the current study is its investigation of the simultaneous impact of the constructs (ILT and PL), which have not been explored with respect to negative affectivity, and, consequently, teacher performance indicators proximate to OCB towards co-workers and students. On the other hand, research shows that female teachers are increasingly likely to engage in OCB towards students and co-workers due to their inherent nature of being helpful as well as being interpersonally and relationally oriented [34]. Male teachers, on the contrary, take an interest only in OCB related to civic virtues [26] whilst being goal-oriented and providing constructive suggestions. But, to date, no study has looked into the gender moderating role in the ILT-NA-OCB towards students and co-workers; hence, this should be further explored. The relationships identified in this study are displayed in Figure 1 below.

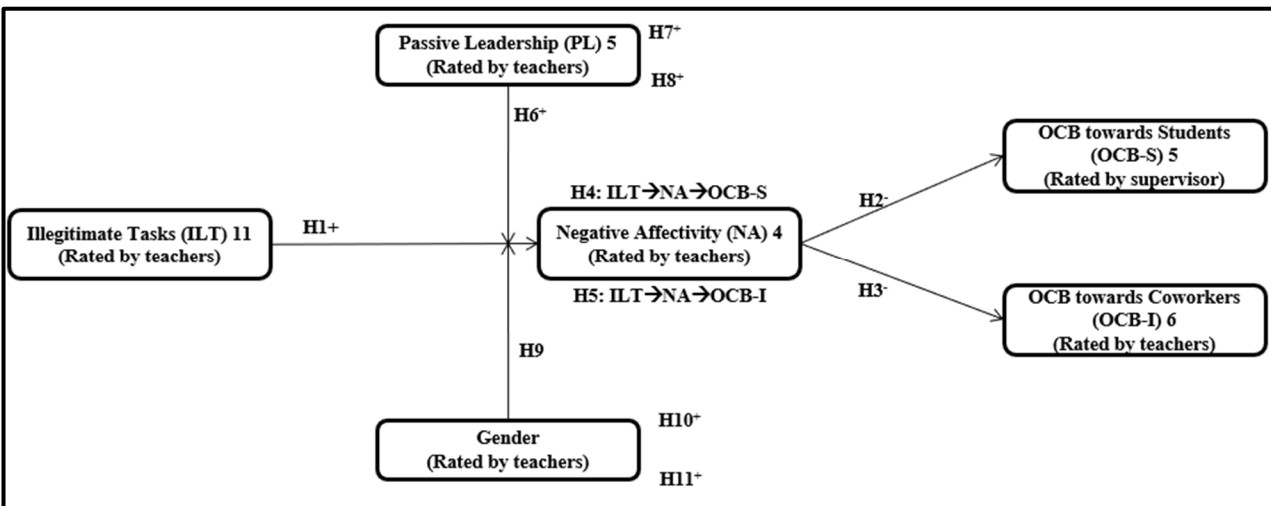

**Figure 1.** The Conceptual Study Model.

In summary, the research aims to answer the following research questions:

1.  How do improper task assignments affect teachers' performance, proxied by OCB?
2.  What is the role of negative affectivity as a connecting mechanism between illegitimate tasks and OCB?
3.  What role does passive leadership play in the illegitimate tasks–negative affectivity relationship?
4.  Does gender have any effect on the illegitimate tasks–negative affectivity–OCB nexus?

The study has several implications for behavioral research. It mainly bridges the evident associated gap in research by examining how illegitimate tasks are perceived, responded to, and coped with by teachers in the education sector. Secondly, it is the first study to analyze the complex pathway of the ILT-NA-OCB nexus with the moderating effects of PL and gender in the education sector, especially targeting the teachers who provide unparalleled support to schools' operations.

## 2. Theoretical Underpinnings, Literature Review, and Hypotheses Development

### 2.1. Illegitimate Tasks

Although no studies to date have identified the ILT-NA nexus, job stressors are posited to have a positive association with burnout and resentment [2,11,35], discrete emotions and fatigue [36], low end-of-day self-esteem [21], turnover intentions and counterproductive work behavior against supervisors and colleagues [10], all of which constitute as proxies of NA. An important study conducted in 2012 showed that a quarter of trainee teachers in a German study reported poor mental health, and 50% of teachers reported poor mental health [37]. These trainees performed the same duties as full-time teachers who joined them on an equal basis. Teachers' trainees who must drudge through central examination, supervision, conferences, and additional duties highlighted their unclear role, which is either the role of trainees or teachers. Such illegitimate tasks were not in the core role of trainees. In parallel, [14] observed emotional depletion as well among the studied sample following a 9-month observation period, once experiencing ILT. Similarly, [38] reported a build-up of emotional fatigue in three burnout aspects (emotional fatigue, depersonalization, and demolished career achievement) in a Queensland sample study of teachers. Thus, ILT assigned beneath or beyond teachers' professional capacity may be degrading and reduce the confidence of a teacher, producing stress and leading to NA.

Therefore, we posit that illegitimate tasks lead to psychological imbalances that produce a host of deleterious effects on teachers' well-being whilst aggravating negativity.

**Hypothesis 1.** *ILTs have a positive relationship with NA.*

### 2.2. Negative Affectivity

In the workplace, illegitimate tasks are often common. Nevertheless, it is not quite clear why teachers react differently, both emotionally and behaviorally, when given the same set of illegitimate assignments [17]. NA is perceived as the extent to which an individual experiences varying levels of conflicting emotions such as agitation, dissatisfaction, pessimism, depression, distress [17,39], avoidance and dwelling on past mistakes, threats and shortcomings, turnover intentions and guilt, and fear and anxiety [39]. Looking into the negative consequences, teachers undergoing negative affectivity are unlikely to provide co-worker support, encounter interpersonal conflicts and increased workload, show absenteeism and tardiness, and exhibit dwindling efficacy, eventually leading to lower performance [2,17]. As such, teachers experiencing NA are unlikely to engage in OCB [24]. OCB towards co-workers includes typical behaviors such as altruism, cooperation, courtesy, a helpful nature, and peacekeeping with fellow workers [11], while OCB towards students comprises the rapid and efficient provision of students' services, students' empathy, and courteous behavior [2]. The NA-OCB association is also underpinned by the COR [40] theory, stating that people are interested in conserving and obtaining materials. The loss and depletion of which cripple them psychologically, thereby dissuading them from engaging

in OCB towards a student\co-worker. The COR theory provides a theoretical justification to explain the mediating impact of NA between ILT and OCB dimensions in the sense that teachers are less oriented towards OCB dimensions as ILT consumes their resources such as time, energy, and positivity, thus amplifying negative affectivity, which in turn leads to demotivation to engage in OCB dimensions. Prior research has frequently employed NA as a mediating/moderating variable [39,41] to study their impact on employee performance, strains, and OCB. Therefore, looking into the negative affectivity–OCB nexus, it can be validated that negative affectivity within individuals leads to negative dispositions at the workplace and forces them to dissuade themselves from OCB towards students\co-workers due to the emotional consequences, encompassing depression, workload, dissatisfaction, demotivation, and role ambiguity, mounting within. Therefore, the paucity of research on the IL-NA-OCB nexus, along with the arguments from the contemporary literature [42], suggests that NA serves as a mediator between IL and OCB dimensions, leading to the following posited hypotheses:

**Hypothesis 2.** *NA is negatively related to OCB towards students.*

**Hypothesis 3.** *NA is negatively related to OCB towards coworkers.*

**Hypothesis 4.** *NA mediates the role between ILTs and OCB towards students, the negative relationship being stronger for teachers experiencing ILTs.*

**Hypothesis 5.** *NA mediates the role between ILTs and OCB towards coworkers, the negative relationship being stronger for teachers experiencing ILTs.*

### 2.3. Passive Leadership as a Moderator

Leadership plays a pivotal role in the impact of workplace stress on teachers' psychological well-being [43,44] and, in turn, the sense of work meaningfulness [45,46]. While ineffective and negative leadership can lead to stress and teacher burnout, effective leadership can foster a positive and encouraging work environment that supports teachers to thrive and excel [45]. With PL, management postpones or neglects any reaction until an issue escalates prior to interfering. This type of leadership strengthens the negative impact of ILT (proxied by role overload, role ambiguity, work pressure, etc.) on NA (proxied by stress, distrust, dissatisfaction, and disempowerment, amongst others), thereby contributing to less OCB towards students and co-workers. The proposed moderating role of PL on the IL-NA-OCB is underpinned both by the COR theory [40] and the SOS theory [29]. The moderating role of PL aggravates the negative IL-OCB relationship, as teachers are devoid of social resources (COR theory) in the form of support and appreciation for handling conflicting and undesirable work situations. Additionally, teachers, when faced with illegitimate tasks, approach leaders for feedback and advice. The prevalence of PL fails to provide solutions for grievances voiced and builds NA. Although few studies have surfaced with PL as a moderator in strengthening the negative effects of IL on work–life balance [47–49], there have been no investigations to date related to the contribution of the moderating effect of PL on the IL-NA-OCB dimension nexus. Therefore, the following hypotheses are posited:

**Hypothesis 6.** *PL moderates the link between ILTs and NA, the relationship being stronger with high PL.*

**Hypothesis 7.** *PL moderates the secondary impact of ILTs on OCB towards students through NA, the mediated effect being stronger with high PL.*

**Hypothesis 8.** *PL moderates the indirect effect of ILTs on OCB towards coworkers through NA, and the mediated effect is amplified with higher PL.*

### 2.4. Gender as a Moderator

Prior research [50,51] has emphasized the moderating role of gender on various aspects of organizational behavior. Women tend to adhere to gender congruent roles, thereby showing a passive acceptance and tolerance [52] towards ILT rather than considering it as non-threatening [52]. At the same time, their counterparts resist ILT, presenting it as a violation of one's professional obligations that is also inconsistent with male gender aspirations.

Limited research explored the moderating role of gender on various OCB-related relationships, such as illegitimate tasks and job fulfillment/internal motivation [52], workplace loneliness, and OCB [51], amongst others. We argue that female teachers are confident multitaskers and proficient in juggling their careers with family lives, leading to passive reactions towards ILT, translating into lower NA, thereby having the resources and altruistic mentality to engage in OCB towards students and co-workers. The following hypotheses are formulated:

**Hypothesis 9.** *The positive relationship between ILT and NA is moderated by gender, with male teachers having a stronger relationship.*

**Hypothesis 10.** *Gender moderates the indirect relationship between ILT and OCB towards students via negative affectivity, such that the indirect connection will be more powerful for male teachers.*

**Hypothesis 11.** *The indirect relationship between ILTs and OCB towards colleagues via NA will be moderated by gender, with male teachers having a stronger indirect relationship than female teachers.*

## 3. Research Materials and Methods

### 3.1. Sample and Procedure

This section presents the collection of data, instrumentation, survey approach, research sample, and the methods adopted to analyze the data. Data were gathered from full-time instructors employed in various subject departments (Math, English, Computer Science, etc.) at private schools in the three largest emirates of the UAE (Dubai, Sharjah, Ajman). A convenience sampling approach was adopted, where schools were chosen on the basis of their accessibility [1]. Convenience sampling was employed to contact the subjects, as the participating schools were identified and were predominantly reached through personal and professional connections. Prior research has emphasized the suitability of convenience sampling when researchers choose to involve individuals who are readily accessible and can conveniently contribute to and participate in a study [53]. Nevertheless, the literature has recognized that convenience sampling is both cost-effective and easily administered (e.g., [54]). This sampling method has been widely utilized in the fields of social science and organizational behavior (e.g., [33,55]).

Data were collected through online surveys shared with teachers and direct supervisors with complete adherence to ethical guidelines. To optimize quality of findings by minimizing the risk of self-reports and reducing the likelihood of methodological bias [25], surveys were administered to supervisors and their direct subordinates in a time-lagged design study. A pilot study was carried out to empirically examine and validate the reliability of developed questionnaires [56]. The teachers would rate their supervisor on aspects related to passive leadership, negative affectivity, etc., and the supervisors rated teachers' service performance. Codes were assigned to questionnaires to match teachers' responses with their line managers' evaluations. Data were collected from the participants in a time-lagged design at three time points.

In the first step, the supervisor assisted us in finalizing teams comprising at least 4 members. Hence, 127 supervisors and 392 directly reporting teachers were approached to participate in the study. In the next step, at Time 1 (T1), these 392 teachers identified in the initial step to complete the demographics questionnaire and rate their perceived illegitimate tasks (i.e., ILT) and their perceptions towards passive leadership practices of their supervisors (i.e., PL) (T1 and Level 1 measures). Of the 392 teachers, 335 com-

pleted and returned the questionnaire, with 85% response rate. Two weeks after the first survey, we approached the 335 teachers of T1 and requested their completion of the teacher-related questionnaire and that they rate their negative affectivity and OCB towards co-workers (i.e., NA, OCB-I) (T2 and Level 1 measures). Of the earlier-identified 335 teachers, 315 completed and returned the questionnaires with a 94% response rate. Two weeks later, we requested that the participating supervisors complete a supervisor-related questionnaire containing OCB-S measures to evaluate their subordinates on service performance (T3 and Level 2 measures), where 110 supervisors responded with an 87% response rate. In summary, from the initially contacted sample of 127 supervisors and 392 subordinates, dyadic data from 100 supervisors and 315 teachers were matched using identifying codes, thus ensuring a 1:3 supervisor–employee ratio. Descriptive statistics of the research sample of teachers (n = 315) revealed that 83% were females and 17% were males, majority (54.5%) were within 23–32 age group (18% were above 32), 73.1% possessed up to ten years of expertise, and 15.7% had more than ten years of experience.

### 3.2. Measures

We adopted validated instruments to measure the important variables. All key constructs were measured using multi-item scale from well-known and reputable instruments in the extant literature. The items were rated on a 5-point Likert scale from "never" to "frequently". For all of the constructs, the scale's construct reliability was reported to be higher than 0.7. The key constructs' measurement is detailed below.

### 3.2.1. ILT (Rated by Teachers)

An 8-item scale created by Semmer et al. [30] is employed in this study to assess the perceived unjustified tasks that teachers encounter. The options for eliciting responses range from 1 (Never) to 5 (Frequently). "Do you have work tasks to take care of, which keep you wondering if they have been done at all?" was one example of a scale item. (Also adopted by [5]).

### 3.2.2. NA (Rated by Teachers)

A 4-item scale is applied [49]. On each item, participants were asked to rate their level of agreement (from strongly disagree to strongly agree). Sample scale items comprised "My job makes me dissatisfied" (also adopted by [57]).

### 3.2.3. OCB (OCB-I) towards Coworkers (Rated by Teachers)

We applied a 6-item scale that was derived from [58]. "I help my coworkers when their workload is heavy" constituted one of the sample scale items (refer to [36]). On each scale item, participants were prompted to indicate their level of agreement or disagreement (from strongly disagree to strongly agree).

### 3.2.4. OCB (OCB-S) towards Students (Rated by Supervisor)

The study utilized a scale with five items extracted from [59]. Participants indicated their agreement or disagreement level (strong disagreement to strong agreement) on each scale item. Sample scale items comprised "This teacher remains after school hours to assist students with content and learning materials" (also adopted by [60]).

### 3.2.5. PL (Rated by Teachers)

The perception of teachers as subordinates towards their leadership was examined through a five-item Likert scale adopted from [61]. Participants rated how strongly they agreed or disagreed with each statement, ranging from strongly disagree to strongly agree. Items from the sample scale included "When teachers need assistance with a problem, my supervisor is not available". This construct, in addition to the subsequent scale items, was validated using numerous recent empirical studies (also adopted by [36]).

3.2.6. Control Variables

Age, gender, and work history were chosen as the control variables because they could affect the constructs used [9].

*3.3. Analytical Techniques*

In this study, correlations between all of the study's variables were examined, and participant demographic features were descriptively analyzed using SPSS 26. A confirmatory factor analysis (CFA) was performed using AMOS 20 to evaluate the construct validity of the measurements. Harman's single-component test was also employed in the study to evaluate whether common method variance was considered a significant problem. One factor model explained 33.642% of the total variation, according to the results of Harman's single-factor test, indicating that common method variance was not a significant issue [62]. We needed to use hierarchical or cross-level techniques for our hypothesis testing because each participant provided data at two levels: the supervisor level (OCB towards students, OCB-S) and the teacher level (Gender, Illegitimate Tasks, PL, NA, OCB towards coworkers, OCB-I). Since linear regression modeling can resolve no independence problems and estimate the impacts of factors at different levels simultaneously, hierarchical linear modeling (HLM) was employed as an analytic technique to examine the hypotheses pertinent to causality and moderation (i.e., Hypotheses H1 through H9). We relied on the recommendations of [63] to examine the moderated mediated conceptual study model by conducting two types of analyses. In particular, the PROCESS Macro Model#4 analysis [63] was used to examine the mediation aspects (i.e., Hypotheses H4 and H5). Thereafter, the moderator was incorporated into the full model via PROCESS macro-Model#10 analysis to measure the whole moderated–mediated conceptual model (i.e., Hypotheses H7, H8, H10, and H11). The aforementioned models were particularly examined via the bootstrapping technique, including 5000 sample size [63]. Significant findings were found by looking at the 95% confidence interval that came from the bootstrapping mediation analysis.

**4. Results**

*4.1. CFA Findings*

Table 1 displays the convergent validity findings in response to [56]. In view of low factor loadings, items ILT1, ILT2, ILT3, and I5 were eliminated from the model [4,64]. Additionally, PL2 was eliminated because of its high error covariance in contrast to the errors of other items. The remaining 26 items had factor loadings that were high, ranging from 0.799 to 0.912 [64].

A satisfactory convergent validity was derived based on the results of the average variance extracted (AVE), Composite Reliability, and Cronbach alpha values. Furthermore, Table 2 below demonstrates that the inter-construct correlations for all constructs are lower than the square root of the average variance extracted for each construct [56].

The correlations across constructs were below the threshold of 0.85, demonstrating acceptable discriminant validity across the variables and model [4]. Females have a significantly higher perception towards OCB-S than males. ILT revealed a significant positive effect on NA but a significant negative effect on OCB towards OCB-I and OCB-S. The correlation between NA and PL was significantly positive, but there was a significance in OCB towards OCB-I and OCB-S. The correlation between OCB-I and OCB-S was found to be significant and positive. Additionally, the constructs NA and OCB-S had the lowest and highest mean values. Findings demonstrated an overall good fit over the four-factor model, and the data gathered $X^2(289) = 352.1$ ($p < 0.01$); $X^2/df = 1.22$, while GFI and AGFI exceeded 0.80 as a threshold. The CFI, comparative fit index, and TLI, Tucker–Lewis index, exceeded 0.90 as a threshold. Similarly, the incremental fit index exceeded 0.9, indicating a good fit [62]. The values of 0.04 for the SRMR and RMSEA, root mean square error, were found to range between 0.03 and 0.08. As such, the derived indices confirm the good fit of the aforementioned model (based on [64,65]).

**Table 1.** Reliability and convergent validity findings.

| Variable | Scale Item | Loadings | AVE | CR | Cronbach Alpha |
|---|---|---|---|---|---|
| Teacher Level (n = 134) | | | | | |
| ILT | ILT 1 | 0.016 [a] | 0.718 | 0.953 | 0.953 |
| | ILT 2 | −0.04 [a] | | | |
| | ILT 3 | −0.017 [a] | | | |
| | ILT 4 | 0.845 | | | |
| | ILT 5 | 0.856 | | | |
| | ILT 6 | 0.824 | | | |
| | ILT 7 | 0.857 | | | |
| | ILT 8 | 0.849 | | | |
| | ILT 9 | 0.846 | | | |
| | ILT 10 | 0.852 | | | |
| | ILT 11 | 0.85 | | | |
| PL [a] | PL 1 | 0.864 | 0.761 | 0.927 | 0.926 |
| | PL 2 | 0.892 [b] | | | |
| | PL 3 | 0.835 | | | |
| | PL 4 | 0.876 | | | |
| | PL 5 | 0.912 | | | |
| NA | NA 1 | 0.852 | 0.692 | 0.900 | 0.899 |
| | NA 2 | 0.799 | | | |
| | NA 3 | 0.845 | | | |
| | NA 4 | 0.83 | | | |
| OCB-I | I 1 | 0.84 | 0.744 | 0.935 | 0.935 |
| | I 2 | 0.853 | | | |
| | I 3 | 0.863 | | | |
| | I 4 | 0.879 | | | |
| | I 5 | 0.133 [a] | | | |
| | I 6 | 0.876 | | | |
| Supervisor Level (n = 100) | | | | | |
| OCB-S | S 1 | 0.846 | 0.739 | 0.934 | 0.934 |
| | S 2 | 0.852 | | | |
| | S 3 | 0.838 | | | |
| | S 4 | 0.87 | | | |
| | S 5 | 0.892 | | | |

[a] Removed from the conceptual model; [b] Removed from the conceptual model.

*4.2. Findings on Hypotheses–HLM*

Regarding HLM analysis, to estimate the supervisor within-group and between-group variability, an unconditional, "intercept only" model for NA, OCB-S, and OCB-I is used. Interestingly, significant within-group variability in supervisors is depicted. Significant results regarding between-group variances in supervisors are derived. NA, OCB-S, and OCB-I have intraclass correlation coefficients (ICC) that are higher than the cutoff of 0.05 [66]. Particularly, between supervisor groups, there is a total of 81.4% variation in NA, 97.7% variation in OCB-S, and 85% variation in OCB-I. The significance of between- and within-group variability is highly important, as it proposes that supervisor-related variables could contribute to the explanation of variation in supervisors' levels of negative affectivity NA, OCB-S, and OCB-I. In other words, these variances show the nested form of the data which supports the adoption of multilevel analysis.

Model 2 has a significantly better fit of the data than Model 1, while Model 3 has a better fit compared to Model 1, which is based on the Chi-square tests using deviance values. Pseudo R2 values support all models' validity. The findings from utilizing HLM to examine the causative and moderation hypotheses are demonstrated in Table 3.

**Table 2.** Correlations, discriminant validity, means, and Standard Deviations.

| | Variable | Mean | SD | 1 | 2 | 3 | 4 | 5 | 6 |
|---|---|---|---|---|---|---|---|---|---|
| 1 | Gender [a] | 1.410 | 0.494 | (1) | | | | | |
| 2 | ILT | 2.469 | 0.816 | −0.069 | (0.847) | | | | |
| 3 | PL | 2.276 | 0.978 | 0.039 | −0.003 | (0.872) | | | |
| 4 | NA | 2.066 | 0.813 | 0.022 | 0.467 *** | 0.404 *** | (0.832) | | |
| 5 | OCB-I | 3.697 | 0.902 | 0.119 | −0.428 *** | −0.118 | −0.479 *** | (0.862) | |
| 6 | OCB-S | 3.805 | 0.864 | 0.193 * | −0.387 *** | −0.05 | −0.427 *** | 0.524 *** | (0.86) |

Note: The scores in parentheses indicate the square root of AVE; The SD represents Standard Deviation; A 5-point Liker scale is applied: Strongly Disagree' is represented by 1 and 'Strongly Agree' by 5; [a]: Two groups were assigned for Gender: 1 for Males, 2 for Females; The Correlations are disclosed based on * $p < 0.05$; *** $p < 0.001$.

Table 3 below and Figure 2 demonstrate a significant positive correlation between ILT and NA, while NA significantly correlates with OCB-S and OCB-I, confirming hypotheses $H1^{+}$, $H2^{-}$, and $H3^{-}$. The interaction terms of ILT × Gender in multiple linear regression analysis was significant in predicting NA; hence, H9 is rejected. Similarly, the interaction effect of ILT × PL was significantly positive in predicting NA, supporting H6.

**Table 3.** Regression analysis results showing mediation and moderation model.

| Variables | NA | | | OCB-S | | OCB-I | |
|---|---|---|---|---|---|---|---|
| | Model 1 | 2 | 3 | 1 | 2 | 1 | 2 |
| Step 0: No Predictor Variable | | | | | | | |
| Intercept | 2.063 (0.082) *** | 0.102 (0.275) | 1.845 ** (0.612) | 3.823 *** (0.081) | 4.114 *** (0.131) | 3.735 *** (0.086) | 4.588 *** (0.184) |
| Step 1: Independent Variables | | | | | | | |
| ILT | | 0.445 ***[H1] (0.074) | −0.292 (0.246) | | | | |
| PL | | 0.335 *** (0.064) | −0.257 (0.167) | | | | |
| Gender | | 0.061 (0.096) | −0.009 (0.279) | | | | |
| NA | | | | | −0.141 **[H2] (0.051) | | −0.414 ***[H3] (0.081) |
| Step 2: Interaction Terms | | | | | | | |
| PL × ILT | | | 0.245 ***[H6] (0.063) | | | | |
| Gender × ILT | | | 0.041 [H9] (0.111) | | | | |
| Model Fit | | | | | | | |
| $\sigma^2$ | 0.128 *** | 0.097 *** | 0.073 *** | 0.015 *** | 0.016 *** | 0.113 *** | 0.133 *** |
| $\tau$ | 0.560 *** | 0.345 *** | 0.349 *** | 0.641 *** | 0.583 *** | 0.641 *** | 0.449 *** |
| $\rho$ | 0.814 | 0.780 | 0.827 | 0.977 | 0.973 | 0.850 | 0.771 |
| Deviance | 289.109 | 234.645 | 221.164 | 213.535 | 206.443 | 294.427 | 273.094 |
| Δ deviance | | −54.464 *** | −13.481 *** | | −7.092 ** | | −21.333 *** |
| Pseudo R$^2$ | | 0.242 | 0.247 | | 0.062 | | 0.150 |

N = 100 supervisors; N = 134 teachers; the unstandardized regressions coefficients are determined from the HLM analysis; the standard error from HLM are the values in parentheses; ** $p < 0.01$. *** $p < 0.001$ (two-tailed); $\sigma^2$ represents the variance within groups ($\sigma^2_w$); $\tau$ represents the variance between groups ($\sigma^2_B$); $\rho$ represents the intra-class correlation coefficient; −2 log likelihood function of the full maximum-likelihood estimate (that determines the model fit) is the deviance measure.

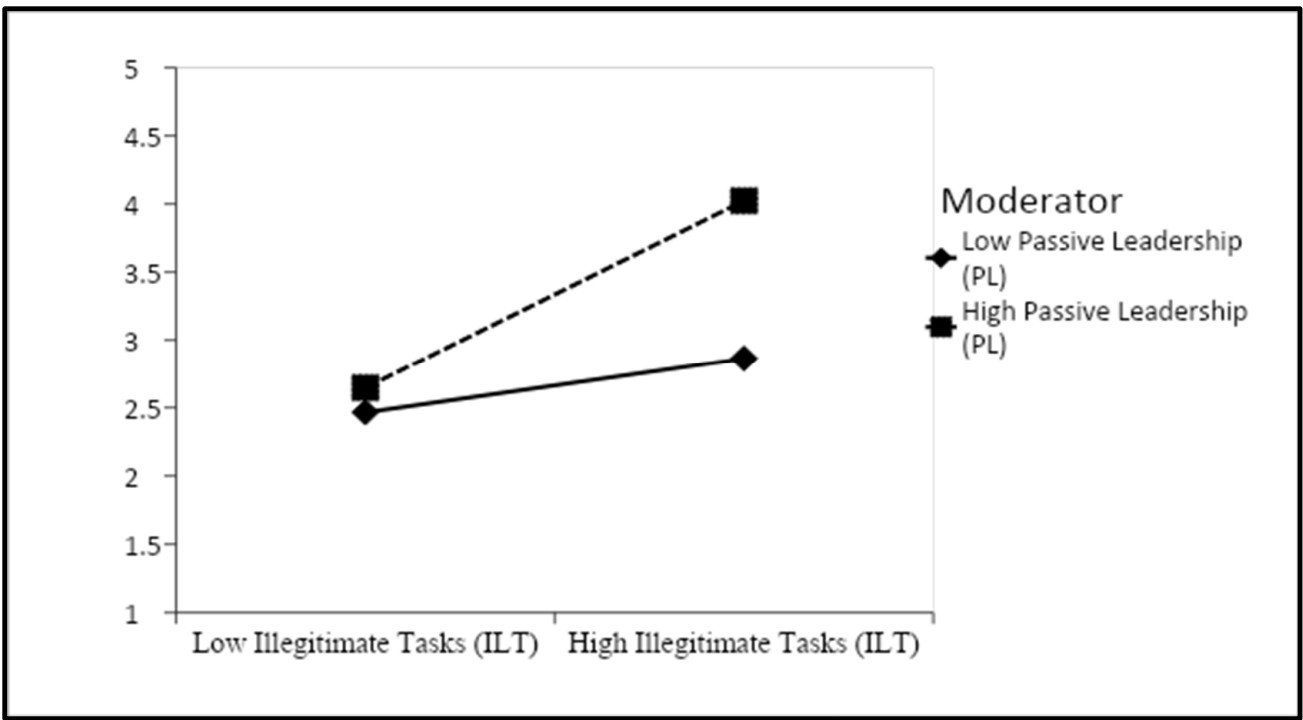

**Figure 2.** PL Moderation effect on the association between ILT and NA.

The resultant interaction, shown in the figure below, demonstrates that ILT increases NA to a larger extent in the event of high, as compared to low, PL, supporting H6 on the positive moderating function of PL for the positive association ILT-NA.

### 4.3. Hypotheses Testing Using Hayes' PROCESS

The findings from utilizing Hayes' PROCESS to examine the mediation (H4 and H5) and moderated mediation (H7, H8, H10, and H11) hypotheses are shown in Table 4. Regarding H7 and H10, bootstrapping results demonstrate that the conditional and indirect relationships are all significant, particularly negative and strong for males in the event of high passive leadership, negative and strong for females in the event of high passive leadership, negative and moderate for males in the event of medium passive leadership, negatively significant and moderate for females in the event of medium passive leadership, negatively insignificant and mild for males in the event of low passive leadership, and generally negative and mild for females.

The results demonstrate that NA mediates the adverse effect of ILT on OCB-S and OCB-I, supporting H4 and H5. A negative and significant indirect effect of ILT on OCB-S through NA is confirmed by bootstrapping estimation. The indirect relationship between ILT and OCB-I by means of NA is similarly negative and significant. According to the index of moderated mediation, the six multi-conditional impacts are significantly distinct from one another for PL's moderated mediation influence, whereas they are not significantly different from one another for the gender's moderated mediation impact. As a result, while hypothesis H10 is rejected, hypothesis H7 is confirmed. The bootstrapping results for the two hypotheses (H8 and H11) show that hypothesis H8 is accepted, whereas hypothesis H11 has been rejected, as depicted in Figure 3. Further information regarding the measurement model using AMOS could be found in Appendix A.

**Table 4.** Findings on mediation and moderated mediation effects via PROCESS.

| Study Model | β | SE | LL95%CI | UL95%CI |
|---|---|---|---|---|
| Mediation Effects via PROCESS macro (i.e., Model 4) | | | | |
| Results from the Bootstrap Analysis of ILT's indirect impact on OCB-S through negative affectivity | | | | |
| ILT → negative affectivity → OCB-S | −0.147 *[H4] | 0.058 | −0.250 | −0.022 |
| Bootstrapping findings for ILT's indirect impact on OCB-I through negative affectivity | | | | |
| ILT → negative affectivity → OCB-I | −0.167 *[H5] | 0.064 | −0.305 | −0.051 |
| Moderated Mediation Analysis using PROCESS macro (model 10) | | | | |
| Bootstrapping analysis of ILT's conditional and indirect impact on OCB-S via multiple conditional levels of negative affectivity, passive leadership, and gender mechanism | | | | |
| Low PL (−1SD = −0.976); Female (−1SD = −0.410) | −0.083 | 0.061 | −0.227 | 0.007 |
| Low PL (−1SD = −0.976); Male (+1SD = −0.590) | −0.063 | 0.058 | −0.191 | 0.033 |
| Medium PL (0SD = 0.000); Female (−1SD = −0.410) | −0.159 * | 0.069 | −0.294 | −0.022 |
| Medium PL (0SD = 0.000); Male (+1SD = −0.590) | −0.139 * | 0.066 | −0.262 | −0.007 |
| High PL (−1SD = 0.977); Female (−1SD = −0.410) | −0.235 * | 0.091 | −0.390 | −0.036 |
| High PL (−1SD = 0.977); Male (+1SD = −0.590) | −0.215 * | 0.088 | −0.367 | −0.023 |
| Index of moderated mediation impact of PL | −0.078 *[H8] | 0.035 | −0.138 | −0.003 |
| Index of moderated mediation impact of Gender | 0.020 [H10] | 0.067 | −0.102 | 0.174 |
| Bootstrapping analysis for conditional indirect effect of the relationship of ILT with OCB-I via NA, PL, and gender mechanism | | | | |
| Low PL (−1SD = −0.976); Female (−1SD = −0.410) | −0.071 | 0.053 | −0.200 | 0.003 |
| Low PL (−1SD = −0.976); Male (+1SD = −0.590) | −0.054 | 0.056 | −0.197 | 0.021 |
| Medium PL (0SD = 0.000); Female (−1SD = −0.410) | −0.136 * | 0.063 | −0.264 | −0.019 |
| Medium PL (0SD = 0.000); Male (+1SD = −0.590) | −0.119 * | 0.068 | −0.271 | −0.007 |
| High PL (−1SD = 0.977); Female (−1SD = −0.410) | −0.201 * | 0.085 | −0.364 | −0.01 |
| High PL (−1SD = 0.977); Male (+1SD = −0.590) | −0.184 * | 0.091 | −0.370 | −0.019 |
| Index of PL moderation effects in mediation | −0.067 *[H9] | 0.033 | −0.131 | −0.002 |
| Index of gender moderation effects in mediation | 0.017 [H11] | 0.056 | −0.105 | 0.133 |

Note: β = standardized effect; SE = standard error for standardized beta; N = 100 supervisors); N = 134 teachers; * $p < 0.05$ (two-tailed).

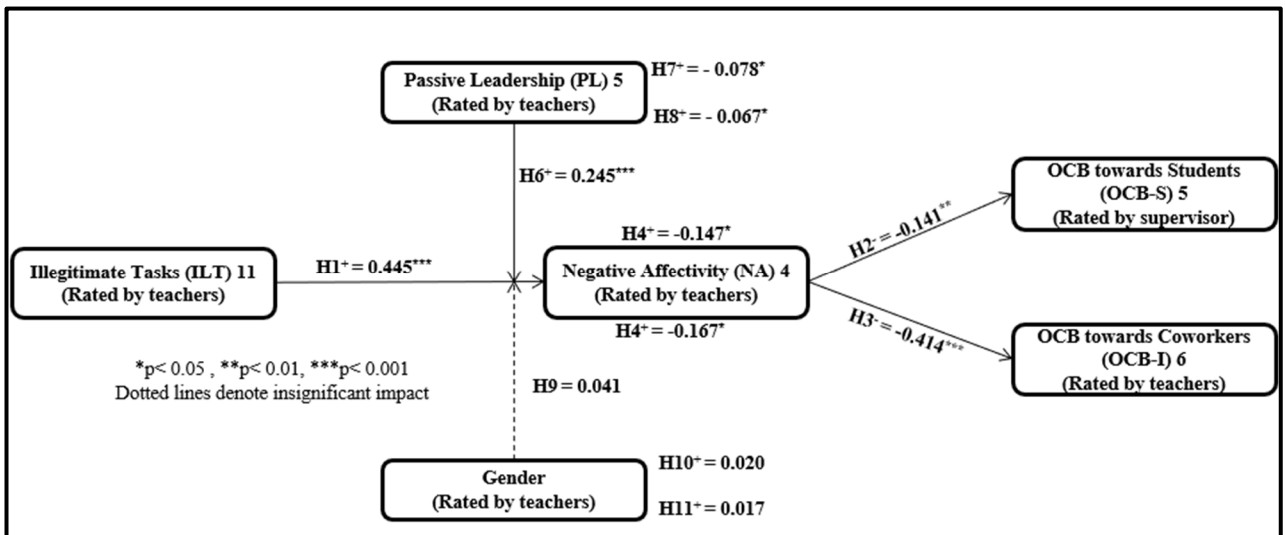

**Figure 3.** Model and estimation findings.

## 5. Discussion and Conclusions

The study constructs ILT, NA, and OCB have been garnering attention in the past decade both in terms of their detrimental effects as well as their dynamic interrelation. The COR and SOS theories form the fundamental underpinning for these organizational stressors and behavior to strengthen the justifications. Addressing the evident gap in the behavioral social science literature, this study attests to the negative impact of illegitimate tasks on OCB towards customers and co-workers, with negative affectivity emerging as a significant factor mediating the aforementioned relationship in the education sector.

The findings evince the acceptance of H1 that illegitimate tasks have a direct impact on negative affectivity, thereby establishing that unreasonable and unnecessary tasks (ILTs) pave the way for the deterioration of physical and psychological well-being [2,67], which

in turn manifests in the form of depression, rapid turnover, increased stress, anxiety [49], and work-related disputes and arguments amongst others. The results conform to the SOS theory [29] and reiterates the fact that in today's competitive world, especially during the trying times of COVID-19, where massive layoffs to curtail costs are evident, illegitimate tasks lead to both mental and ethical dilemma among academic staff thereby arising varying levels of fear, resentment, self-denial, waning motivational levels and decreased confidence levels, all of which are significant outcomes of negative affectivity.

The indirect impact of NA on OCB towards students/co-workers, as well as the mediating effects of NA between ILT and OCB, shows a parity to the key findings of [10] that negative affectivity leads to counterproductive behavior both at the workplace and with colleagues. The authors attribute this to the detrimental effects of negativity, which weigh heavily on one's mindset, thus dissuading one from engaging oneself with students, parents, colleagues, and the school altogether. These behaviors can stem from the burnout, demotivation, stress, and agitation inherent in negative affectivity, which later on integrate into one's attitude and manifest in the form of strained relations with peers and students.

The findings also align with the COR theory, except in that they show agreement with the findings of [42]. As a solution, the authors suggest more social exchanges and interactions among employees as a self-initiative to mitigate stress levels, thereby enhancing OCB towards students and co-workers.

Concurrently, results support H6–H8, all of which signal the detrimental effects of the moderator passive leadership on negative affectivity as well as OCB towards students and co-workers. The findings lend support to the COR theory [40] that individuals are keen on preserving their emotional, physical, and social resources in order to acquire other intangible resources like appreciation and acknowledgment. However, once they notice that the efforts are in vain and disregarded and exploited by their supervisors (leaders), they amass psychological and emotional upheavals [61] due to their inability to retaliate, and they gradually dissuade themselves from value-adding activities both to the organization as well as with colleagues [48]. It is to be noted that pandemic-related uncertainties in organizations have put enormous pressure on business leaders to maintain profit levels by minimizing human resources, thereby increasing the pressure on existing employees. These unprecedented transformations have prompted even participative leaders to emulate passive leaders, thereby refraining from providing mental, moral, and leadership support to existing employees apart from turning a deaf ear to their woes and concerns, all of which have aggravated stress levels and demotivated employees to engage in as OCB towards students and co-workers. However, self-motivation, a resilient personality, opportunity seeking, and a positive attitude can help alleviate the detrimental effects of illegitimate tasks, thereby minimizing negative affectivity and enhancing OCB towards students and co-workers. Nevertheless, unwavering leadership support outweighs the benefits of the former traits and is monumental in mitigating negative affectivity caused by illegitimate tasks. Supportive leadership, especially during turbulent phases (such as pandemics), is likely to elicit productive, altruistic, and innovative behaviors [2] as employees develop an urge to reciprocate the support received.

Unexpectedly, the moderating effects of gender show that gender variances do not affect the ILT-NA-OCB relationship. The current study's findings are inconsistent with international findings (e.g., [52]), all of which support the traditional gender ideology of the inherent enduring nature of women and their apathy towards illegitimate tasks. In principle, we argue that gender ideology and culture play a phenomenal role in a teacher's approach and mindset towards challenges and opportunities. However, the sample for this study is extracted from the UAE, a country that advocates encouraging and supportive policies for women's empowerment. Gender equality receives a top priority and is of paramount importance in the UAE [50], where the Constitution ensures that men and women have equal rights. Women under the Constitution are afforded the identical legal status, entitlement to titles, educational access, professional practice rights, and inheritance privileges as men [50], which could further explain the results showing that women and

men develop similar mental and psychological reactions to illegitimate tasks, revealing insignificance in gender-based responses towards illegitimate tasks.

*Managerial Implications*

The results delineate several practical implications. First, this study demonstrates that ILT can be highly costly at the school and individual levels in terms of large teacher and student turnover, for which immediate and adequate managerial intervention strategies are paramount. Job redesigns can help mitigate and evade illegitimate tasks, cultivate regular and ongoing feedback loops, and encourage open communication between managers and subordinates to stay abreast of tasks perceived as illegitimate. While this research has disclosed associated NA as a response to ILT, the administration should consider how their task assignment can either empower or demean teachers.

Additionally, the mediating role of NA presents an essential channel for ILT to exert a negative impact on OCB, which managers work hard to promote. Given that ILTs are occasionally unavoidable, it is crucial that supervisors implement supportive practices in managing teachers' emotions to help them feel better at work and prevent more decline into unfavorable outcomes. In that sense, PD programs and training modules are one way to promote the healthy processing of emotions. For instance, international findings (e.g., [68]) show that teachers' emotional regulation skills and general well-being state improved following affect regulation training, permitting them to better cope with difficult emotions [2]. Managers can pre-acknowledge the unfavorable task inherent in the job assignment and diplomatically detail the requirement to complete the tasks to minimize the potential undesirable state of feelings experienced by the teacher [44,45].

Third, this study shows the critical role played by indifferent leaders in elevating the damaging influences of negative emotions as a result of ILT on OCB. Nevertheless, ILTs are less likely to be totally eradicated from the workplace. Still, supportive school leadership can assist in changing teachers' perceptions of such tasks into more meaningful and worthwhile ones. In the face of the current situation, organizations in the educational sector should encourage engaged and constructive leadership among their managers, including the cultivation of inspirational leadership [61] and the development of strategies and plans that tackle PL practices in addition to offering leadership coaching and training to proactively address passive leadership while progressing towards 360-degree feedback.

## 6. Limitations and Future Directions

While this study has certain limitations, it also brings about exciting avenues for further research. Firstly, data collection focused on teachers working in the UAE's education sector. In that sense, the generalizability of the results towards additional populations and amongst other sectors and industries is limited. Hence, the same set of key constructs and models could be employed in further research studies addressing different countries and sectors. Secondly, while acknowledging that such variables may vary within days, this research study examines the relationship among three key constructs, namely, ILT, NA, and OCB, over a comparatively long time. In that regard, ILTs that teachers encounter on a daily or weekly basis can have various negative effects on OCB over time. The moderating effect of PL, which can reinforce the negative impact of ILTs on OCB through NA, is the study's final finding. A further study may take into account various moderating factors, such as the good and bad sides of leadership traits, such as despotic leaders or the LMX leadership style, and the good and bad sides of teachers' personalities with respect to their capacity for emotional regulation and paranoia. A comparative study between teaching and non-teaching staff to unveil the variations in the impacts of work stressors is another area of research to be uncovered. Finally, a cross-country study can further provide a comprehensive idea of how work stressors affect personalities in different regions.

**Author Contributions:** Methodology, R.A.; Formal analysis, R.P.; Writing—original draft, N.S.; Writing—review & editing, L.M. All authors have read and agreed to the published version of the manuscript.

**Funding:** This research received no external funding.

**Institutional Review Board Statement:** Ethical review and approval were waived for this study on human participants in line with the local legislation and institutional requirements. The survey examined in this work was entirely voluntary and only included adult participants who were informed about the research aims and their right to exit the study whenever they wish, without any repercussions.

**Informed Consent Statement:** The survey examined in this work was entirely voluntary and only included adult participants who were informed about the research aims and their right to exit the study whenever they wish, without any repercussions.

**Data Availability Statement:** The raw data presented in this study are available on request from the corresponding author. The raw data are not publicly available, however will be made available by the authors, without undue reservation.

**Conflicts of Interest:** The authors declare no conflict of interest.

**Appendix A**

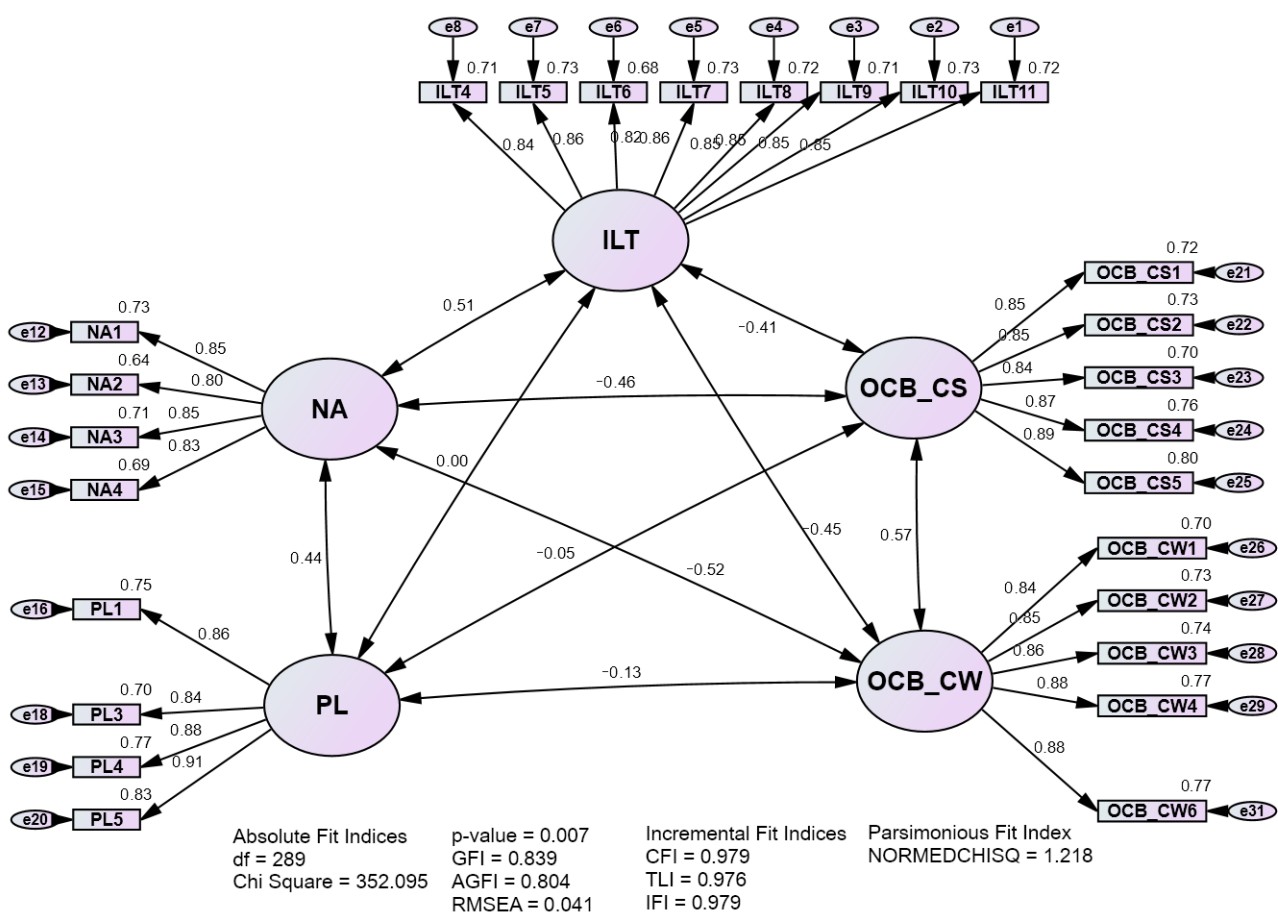

**Figure A1.** AMOS CFA graphs.

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
