# Peer review of "Illegitimate Tasks, Negative Affectivity, and Organizational Citizenship Behavior among Private School Teachers: A Mediated–Moderated Model"

_sustainability, doi:10.3390/su16020733_

Round 1
Reviewer 1 Report
Comments and Suggestions for Authors
The manuscript explores the impact of illegitimate tasks on teachers’ Organizational Citizenship Behavior (OCB) with negative affectivity as a mediator whilst gender and passive leadership act as moderators. Using questionnaires, the authors selected 415 private school teachers as the study participants. The results show that illegitimate tasks have an indirect and statistically significant impact on OCB, the effects of which are mediated by negative affectivity. However, there are still some things that could be improved with the structure and content of the manuscript.
1. I suggest adding a section about Research Questions.
2. It would be better to provide literature support for the criterion of model fit indices in line8 of P15.
3. I suggest adding an introduction of the HLM analysis and Hayes' PROCESS in the 3. Materials and Methods section.
4. As for each scale in 3.2. Measures, please illustrate whether they are all 5-point Likert scales or not.
5. Please provide a further in-depth discussion of the findings. For example, in terms of the statement “The results contradict the findings by Anthony et al. (2016) and Omansky et al., (2016)” in line 19 of P22, the authors can analyze in depth the reasons why the result of the present study is different from those of previous studies.
6. Please double-check the initial capitalization of the headings and spelling of words. For example, the heading “2. Theoretical underpinnings, Literature Review, and Hypotheses formulation” is not uniformly capitalized.
7. Please do a copyediting before resubmitting the revised version.
Comments on the Quality of English LanguageThe manuscript explores the impact of illegitimate tasks on teachers’ Organizational Citizenship Behavior (OCB) with negative affectivity as a mediator whilst gender and passive leadership act as moderators. Using questionnaires, the authors selected 415 private school teachers as the study participants. The results show that illegitimate tasks have an indirect and statistically significant impact on OCB, the effects of which are mediated by negative affectivity. However, there are still some things that could be improved with the structure and content of the manuscript.
1. I suggest adding a section about Research Questions.
2. It would be better to provide literature support for the criterion of model fit indices in line8 of P15.
3. I suggest adding an introduction of the HLM analysis and Hayes' PROCESS in the 3. Materials and Methods section.
4. As for each scale in 3.2. Measures, please illustrate whether they are all 5-point Likert scales or not.
5. Please provide a further in-depth discussion of the findings. For example, in terms of the statement “The results contradict the findings by Anthony et al. (2016) and Omansky et al., (2016)” in line 19 of P22, the authors can analyze in depth the reasons why the result of the present study is different from those of previous studies.
6. Please double-check the initial capitalization of the headings and spelling of words. For example, the heading “2. Theoretical underpinnings, Literature Review, and Hypotheses formulation” is not uniformly capitalized.
7. Please do a copyediting before resubmitting the revised version.
Reviewer 2 Report
Comments and Suggestions for Authors
Thank you for the opportunity to review this study of that investigates the impact of illegitimate tasks on Organizational Citizenship Behavior (OCB) with negative affectivity as a mediator whilst gender and passive leadership acting as moderators in the education sector. Overall I thought the Methods, Results, and Discussion were generally well-written; however, the study has a considerable large scope and is not always easy to follow and monitor the validation of a large number of hypotheses. Much of the theoretical framing in well presented but a huge number of acronyms is also hard to follow.
Reviewer 3 Report
Comments and Suggestions for Authors
Dear Researchers,
We are truly appreciative of the effort you invested in this insightful study on social sustainability and employee satisfaction within private schools in the UAE. The introduction of a sophisticated sequential mediation-moderation model to evaluate the influence of unreasonable tasks on teachers' Organisational Citizenship Behaviour (OCB), with a special focus on negative affectivity, gender, and passive leadership, is commendable.
I have some observations to share that could contribute to strengthening your manuscript:
Sampling Methodology
You've articulated the convenience sampling approach effectively. However, a deeper insight into the reasons for selecting this specific sampling approach, and a discussion on its limitations and potential biases, would enhance the comprehensiveness and credibility of your study.
Measures and Scales
While established scales like ILT, NA, and others have been employed, the adaptation and validation of these scales within the context of the UAE’s unique educational setting are not discussed. A detailed examination of any required adaptations, considering the cultural uniqueness, would add depth and relevance to the findings (Bristol-Rhys, 2010).
Bibliography
It’s paramount to update and expand the bibliography to encompass contemporary studies. Including only one citation from 2022 and omitting references from 2023 might impact the research's relevancy. Incorporating more recent studies will anchor your research in the latest insights, enhancing its validity and pertinence in the ongoing discourse.
Citation Style
The adherence to the numeric citation style is a prerequisite for this journal. Ensuring that each in-text citation correlates with the numbered reference in the bibliography will align your submission with the journal’s standards.
We are looking forward to witnessing the refinement and enrichment of this already significant piece of work. Your contribution to the field is invaluable.
Best wishes,
